# Diversity of Ectomycorrhizal Fungal Communities in Four Types of Stands in *Pinus massoniana* Plantation in the West of China

**Xiangjun Li** [1,2], **Wensi Kang** [1,2], **Size Liu** [1,2], **Haifeng Yin** [1,2], **Qian Lyu** [1,2], **Yu Su** [3], **Junjie Liu** [1,2], **Jiangli Liu** [1,2], **Chuan Fan** [1,2], **Gang Chen** [1,2], **Kuangji Zhao** [1,2] and **Xianwei Li** [1,2,*]

1   College of Forestry, Sichuan Agricultural University, Huimin Road 211, Chengdu 611130, China; lee_sicau@163.com (X.L.); wensikang@126.com (W.K.); size_leo@126.com (S.L.); yhfeng312@163.com (H.Y.); lvqian1109@163.com (Q.L.); liujunjie970310@163.com (J.L.); ljl1004908@163.com (J.L.); fanchuan@sicau.edu.cn (C.F.); g.chen@sicau.edu.cn (G.C.); zhaokj@sicau.edu.cn (K.Z.)
2   Key Laboratory of National Forestry and Prairie Bureau on Forest Resources Conservation and Ecological Security in the Upper Reaches of Yangtze River, Sichuan Agricultrual University, Chengdu 611130, China
3   Sichuan Academy of Forestry, Xinghui Road West, Chengdu 610036, China; yusu110@163.com
*   Correspondence: lxw@sicau.edu.cn

**Abstract:** Ectomycorrhizal (ECM) fungi can form symbioses with plant roots, which play an important role in regulating the rhizosphere microenvironment. As a broad-spectrum ECM tree species, *Pinus massoniana* forms symbiotic relationship called mycorrhiza with various ECM fungal species. In this study, four types of forests were selected from a 38-year-old *Pinus* plantation in eastern Sichuan, namely, pure *P. massoniana* forest (MC), *P. massoniana* mixed with *Cunninghamia lanceolata* forest (MS), *P. massoniana*–*Cryptomeria fortunei* forest (ML), and *P. massoniana*–broadleaved forest (MK), the species mixture ratio of all forests was 1:1. The ITS2 segment of ECM root tip sequenced by high-throughput sequencing using the Illumina MiSeq sequencing platform. (1) The ECM fungi of these four *P. massoniana* forests showed similar dominant genera but different relative abundances in community structure during the three seasons. (2) The alpha diversity index of ECM fungi was significantly influenced by season and forest type. (3) Soil pH, soil organic matter (SOM), total nitrogen (TN), C/N ratio, and total phosphorus (TP) influenced the ECM fungal community structure in different seasons. In summary, there were significant differences in ECM fungal communities among different forest types and different seasons; the colonization rate of ECM fungal in *P. massoniana*–*Cunninghamia lanceolata* was the highest, so we infer that *Cunninghamia lanceolata* is the most suitable tree species for mixed with *P. massoniana* in three mixture forests.

**Keywords:** alpha diversity; community structure; *Cunninghamia lanceolata*; ECM fungi; mixed forests; *P. massoniana*; soil chemical properties

## 1. Introduction

*Pinus massoniana* Lamb. is a native tree species endemic to China that has important economic and ecological value and has symbioses with a variety of fungi that form ectomycorrhizae [1]. Long-term pure forest management leads to a decline in plantation fertility and productivity [2]. Previous studies have shown that the resources of tree species are separated and that the competition among trees is small in mixed forests; thus, mixed forests show higher productivity and biodiversity than pure forests [3]. Mixed forests not only increase the diversity of tree species, but they also change the living environment of the animals, plants, and microorganisms. ECM fungi play an important role in the *P. massoniana* plantation ecosystem [4]; therefore, studying the ECM fungal community structure of *P. massoniana* plantations can provide a theoretical basis for a mechanism of stability and the sustainable development of this forest ecosystem.

ECM fungi promote the growth of host plants. Ectomycorrhizal symbionts mainly expand the absorption area and range of the root system through the structure of the sheath and epitaxial mycelia, thus promoting the absorption of water and nutrients by the host plants [5]. Most of the trees that compose the forest do not have "real" root systems, and ectomycorrhizae are their absorption organs [6]. ECM can improve the drought resistance of host plants. The epitaxial hyphae of ectomycorrhizae increase the water absorption area and improve the hydraulic conductivity of the root system, and the ring traps can prevent water loss [7,8]. When plants are stressed by heavy metals, ectomycorrhizae also play a "barrier" role, which can absorb and resist part of the heavy metal ion damage to plants [9]. Inoculation experiments showed that *P. massoniana* seedlings inoculated with ECM fungi show improved adaptability to heavy metal stress and an improved survival rate compared with non-mycorrhized seedlings [10,11]. The ECM fungi have a rich diversity of species, ecosystems and functions. The huge hyphal network of ECM can almost be connected with various plants in the community for nutrient exchange, energy flow and information transmission [12]. Environmental and biological factors significantly affect fungal communities [13]. It has been reported that soil nutrients are the most important factor affecting the diversity and richness of ECM fungi, and a higher soil nutrient content has a negative impact on the diversity and richness of ECM fungi [14,15].

This study was carried out in a 38-year-old *P. massoniana* plantation in a mountainous area in Huaying city. Four types of forests were selected from the plantation, namely, pure *P. massoniana* plantation (MC), *P. massoniana–Cunninghamia lanceolata* (MS), *P. massoniana–Cryptomeria fortunei* forest (ML), and *P. massoniana–*broadleaved forest (MK), the mixed ratio of all forests was 1:1. This study investigated the differences in ECM diversity and soil chemical properties among the four types of forests to reveal the chemical parameters of soil suitable for the development of ECM community and aimed to provide theoretical support for the cultivation of large-diameter timber of *P. massoniana*. In addition, this study provided a scientific basis for the reconstruction and sustainable development of *P. massoniana* plantations in the low hill districts of the Sichuan Basin, China.

## 2. Materials and Methods

### 2.1. Study Area

The study area is in the Dongfanghong Forest Farm (106°45′–106°47′ E, 30°14′–30°20′ N), Huaying City, Sichuan Province, China, with an altitude of 324–809 m, which is in the low mountainous area of eastern Sichuan. This area is in a humid monsoon climate zone with a mid-subtropical climate, abundant but uneven rainfall, and large annual temperature differences. The soil in this area is yellow earth characterized by low fertility and a lack of phosphorus. The annual average temperature is 17.2 °C, and the average rainfall is 1087.8 mm, the experimental trees were not fertilized, but there are necessary silvicultural measures, such as removing weeds and dead trees. Understory shrubs mainly consist of *Lindera glauca*, *Rubus chroosepalus*, and *Litsea cubeba*. The herb layer mainly consists of *Humata repens*, *Dicranopteris dichotoma*, and *Setaria plicata*.

The *P. massoniana* plantation was established in 1980 (LY/T 2908−2017). Four stand types were selected from the plantation (Table 1), namely, pure *P. massoniana* plantation (MC), *P. massoniana–Cunninghamia lanceolata* (MS), *P. massoniana–Cryptomeria fortunei* forest (ML), and *P. massoniana–*broadleaved forest (MK); the mixture ratio of all forests was 1:1, in which MS and ML were same-age forests, and MK was a mixed alien coniferous and broadleaved forest formed by replanting *Cinnamomum camphora*. The trees of the mixed forest were replanted in 2015 and the same planting density was maintained.

**Table 1.** Overview of the four stand types in the *P. massoniana* plantation.

| | Altitude (m) | Aspect | Breast Diameter (cm) | Height (m) | Crown Density | Type of Mix | Mixture Ratio |
|---|---|---|---|---|---|---|---|
| MC | 690 | Southeast | 17.8 | 13.8 | 0.7 | - | - |
| MS | 776 | South | 18.1 | 14.3 | 0.7 | Interline | 1:1 |
| ML | 630 | South | 18.6 | 14.8 | 0.7 | Interline | 1:1 |
| MK | 647 | South | 18.4 | 14.6 | 0.7 | Interline | 1:1 |

*2.2. Sample Collection and Processing*

Three 20 m × 20 m sampling plots were located in each *P. massoniana* forest in each of the four different stand types, and we chose three trees with the average breast-height diameter in each plot. The distance between any two trees was greater than 10 m to ensure the independence of the mycorrhizal samples [16]. Mycorrhizal samples were collected in April, July, and October 2019 (spring, summer, and autumn, respectively). The same tree was not selected again when sampling in different seasons. Mycorrhizal root tips were collected with a root-cutting knife at a soil depth of 0–30 cm. The samples were placed in an ice box, transported to the laboratory within 48 h, and stored in a refrigerator at 4 °C for no more than a week. Soil was collected at a depth of 0–30 cm from each sample tree from three different directions. Soil samples for physical property analysis were collected with ring knifes, and soil samples for chemical property analysis were collected with a soil drill at a depth of 0–30 cm. Soil samples were air-dried indoors and ground and filtered through a 2 mm sieve for the determination of chemical properties.

Soil pH was measured with an electronic pH meter in a 1:2.5 water and soil suspension after stirring for 10 min with a glass rod followed by standing for 1 h (PHS-25CW, BANTE Instruments Limited, Shanghai, China). The organic matter content was determined by using the dichromate volumetric method/dilution heat method, total soil nitrogen was determined by using the semimicro-Kjeldahl method [17].

For calculation of the ECM fungal colonization rates, root tips were observed by light microscope (DMM-400C) and the number of the total root tips and mycorrhizal root tips were recorded [18].

$$\text{Colonization rate } (\%) = \frac{Sc}{Sr} \times 100\% \tag{1}$$

where $Sc$ is the number of mycorrhizal root tips and $Sr$ is the number of the total root tips.

*2.3. Molecular Identification of Ectomycorrhizal Fungi*

Microbial DNA was extracted using HiPure Soil DNA Kits (or HiPure Stool DNA Kits) (Magen, Guangzhou, China) according to the manufacturer's protocols. The ITS2 segment of ECM was sequenced by high-throughput sequencing using the Illumina MiSeq sequencing platform, RNA gene was amplified by PCR (95 °C for 2 min, followed by 27 cycles at 98 °C for 10 s, 62 °C for 30 s, and 68 °C for 30 s and a final extension at 68 °C for 10 min) using primers ITS3_KYO2 (5′-GATGAAGAACGYAGYRAA-3′) and ITS4 (5′-TCCTCCGCTTATTGATATGC-3′) [19], where the barcode was an eight-base sequence unique to each sample. PCRs were performed in triplicate in a 50 μL mixture containing 5 μL of 10 × KOD Buffer, 5 μL of 2.5 mM dNTPs, 1.5 μL of each primer (5 μM), 1 μL of KOD Polymerase, and 100 ng of template DNA. The 1.2 Illumina HiSeq 2500 sequencing amplicons were extracted from 2% agarose gels and purified using the AxyPrep DNA Gel Extraction Kit (Axygen Biosciences, Union City, CA, USA) according to the manufacturer's instructions and quantified using an ABI StepOnePlus Real-Time PCR System (Life Technologies, Foster City, CA, USA). Purified amplicons were pooled in equimolar amounts and paired-end sequenced (2 × 250) on an Illumina platform according to standard protocols. The raw reads were deposited into the NCBI Sequence Read Archive (SRA) database.

The effective tags were clustered into operational taxonomic units (OTUs) of ≥97% similarity using the UPARSE [20] pipeline. The tag sequence with the highest abundance

was selected as the representative sequence within each cluster. Between groups, Venn analysis was performed in R project (version 3.4.1) to identify unique and common OTUs.

The representative sequences were classified into organisms by a naive Bayesian model using the RDP classifier [21] (version 2.2) based on the SILVA database (https://www.arb-silva.de/ (accessed on 6 May 2021)) or Greengene [22] database, with confidence threshold values ranging from 0.8 to 1. The abundance statistics of each taxonomy were visualized using Krona [23] (version 2.6). Biomarker features in each group were screened by Metastats10 (version 20090414) and LEfSe software [24] (version 1.0).

### 2.4. Analysis of ECM Species Composition

According to the sequence information of OTUs, species annotations were made from four ranks of phylum, class, family and genus, using UPARSE to construct the OTUs. In the process of constructing the OTUs, UPARSE selects representative sequences (the tag sequence with the highest abundance in OTUs) and uses RDPClassifier (v2.2) to set these representative sequences in unit (v2016_11_20_ver7) species annotations and to set the confidence threshold to 0.8. After obtaining the species annotation information of each OTU to study the phylogenetic relationship before the OTUs, KRONA (v2.6) was used to interactively visualize the species annotation results.

In this study, for the ECM of the four forest types in spring, summer and autumn obtained by sequencing, the genera that accounted for $\geq 1.00\%$ of the total abundance of the available annotations were classified as the dominant genera. The dominant genus with a total abundance of $\geq 50.00\%$ of the dominant genera was classified as an absolute dominant genus; and all genera except the dominant genus were classified as rare genera.

### 2.5. Calculations

Based on the OTU abundance results, QIIME was used to calculate the alpha diversity of each sample. Indices such as Shannon (Equation (2)), Simpson (Equation (3)), Chao1 (Equation (4)) and ACE (Equation (5)) were selected to reflect the alpha diversity of the ECM communities. Chao1 and ACE indices were used to predict the types of microorganisms in the sample (the number of OTUs) based on the number of tags and the number of OTUs measured and the relative ratio.

$$H = -\sum_{i=1}^{S} (P_i \ln P_i) \tag{2}$$

$$D = 1 - \sum_{i=1}^{S} P_i^2 \tag{3}$$

$$S_1 = S_{obs} + \frac{F_1^2}{2F_2} \tag{4}$$

$$S_{ace} = S_{abund} + \frac{S_{rare}}{C_{ace}} + \frac{F_1}{C_{ace}} \gamma_{ace}^2 \tag{5}$$

where $P_i$ is the ratio of the OTU abundance of the $i$-th ECM fungal species to the total OTU abundance; $S_{obs}$ is the actual number of OTUs observed in the plot; $F_1$ is the number of species with only 1 in the sample; $S_{abund}$ is the number of species that appeared more than 10 times in the sample; $S_{rare}$ is the number of species that appeared no more than 10 times; and $C_{ace}$ ($C_{ace} = 1 - \frac{F_1}{N_{race}}$) represents the proportion of singletons in all species with a low abundance (occurrence $\leq 10$ times).

### 2.6. Statistical Analysis

Microsoft Excel 2016 and SPSS 25.0 were used for data processing and analysis, and Origin 8 was used to create the figures. One-way analysis of variance (one-way ANOVA) and Tukey's honestly significant difference test (HSD) were used to analyze the significant differences in the ECM diversity index among the four types of forest. Two-way ANOVA

was used to test the influence of stand type, season and their interaction on the ECM diversity index.

Canono5 [25] was used to perform DCA analysis on the ECM species data of the four forest types, and RDA or CCA analysis was chosen according to the size of the first axis of the lengths of gradient in the analysis results. If the value was greater than 4.0, CCA analysis was used; if the value was between 3.0–4.0, both RDA and CCA could be selected; and RDA was chosen if the value was less than 3.0. Canono5 was also used to perform RDA/CCA analysis on the ECM species data with soil physical and chemical properties.

## 3. Results

### 3.1. Composition of ECM Fungi in Four Types of Pinus massoniana Forests

The ECM fungal colonization rates in MC, MS, ML and MK were 54.79%, 60.83%, 58.98% and 54.96%, respectively, in spring (Table 2), and the colonization rate in MS was found to be significantly higher than that in MC and MK ($p < 0.05$). In summer, the infection rates in MC, MS, ML and MK were 56.75%, 63.99%, 59.47% and 53.65%, respectively, among which the colonization rates in MS, ML and MK significantly differed from each other. The ECM fungal colonization rates in autumn were 55.50%, 59.79%, 56.32% and 57.66%, respectively, and the rate in MS was significantly higher than that in MC ($p < 0.05$). The colonization rate of ECM fungi in MS was the highest in the three seasons. Except for MK, the colonization rate of ECM fungi in the other three stand types was the highest in summer.

**Table 2.** The colonization rates of ECM fungi in the four types of *P.massoniana* plantations.

|  | Spring | Summer | Autumn |
|---|---|---|---|
| MC | 54.79 ± 1.07 bA | 56.75 ± 0.23 bcA | 55.50 ± 0.79 bA |
| MS | 60.83 ± 2.17 aA | 63.99 ± 1.16 aA | 59.79 ± 1.01 aA |
| ML | 58.98 ± 1.61 abA | 59.47 ± 1.38 bA | 56.32 ± 1.65 abA |
| MK | 54.96 ± 0.99 bA | 53.65 ± 1.38 cA | 57.66 ± 1.05 abA |

Note: Different lowercase letters indicate significance for colonization rates of ECM fungi in the different stand types in the same season at the $p < 0.05$ level; different uppercase letters indicate significant differences in different seasons in the same stand types at the level of $p < 0.05$; ± followed means of standard error.

In spring, the ECM fungal sequences in the four forest types were divided into 157 OTUs, of which seven OTUs were not identified as genera. All ECM fungi belonging to two phyla, three classes, ten families and 13 genera were identified. In summer, the ECM fungal sequences in the four forest types can be divided into 449 OTUs, of which 34 OTUs have not been identified. In autumn, the ECM sequence in the four types of forest can be divided into 306 OTUs, of which 48 OTUs have not been identified.There are 54 OTUs exist in every forest (Figure 1).

At the phylum rank, the ECM from the four types of forest mainly belonged to Basidiomycota and Ascomycota, of which Basidiomycota was the dominant phylum (Figure 2, Table 3). At the class rank, the ECM fungi were mainly from Agaricomycetes, Dothideomycetes and Pezizomycetes, of which Agaricomycetes was the dominant class. At the family rank, the ECM fungi in MC mainly belonged to nine families, and the dominant families were Russulaceae (74.72%), Thelephoraceae (15.65%), Clavulinaceae (5.52%) and Gloniaceae (1.46%). The ECM fungi in MS comprised seven families, and the dominant families were Russulaceae (71.84%), Clavulinaceae (22.49%), Atheliaceae (3.62%) and Thelephoraceae (1.89%). The number of ECM families in ML was the same as that in MS, but the dominant families were different. Atheliaceae (3.57%), Russulaceae, Thelephoraceae and Clavulinaceae were the dominant families among the four stand types. At the genus rank, *Russula, Tomentella* and *Clavulina* were the dominant genera in the four forest types. *Tylospora* was the dominant genus in ML, and *Wilcoxina* was the dominant genus only in MK.

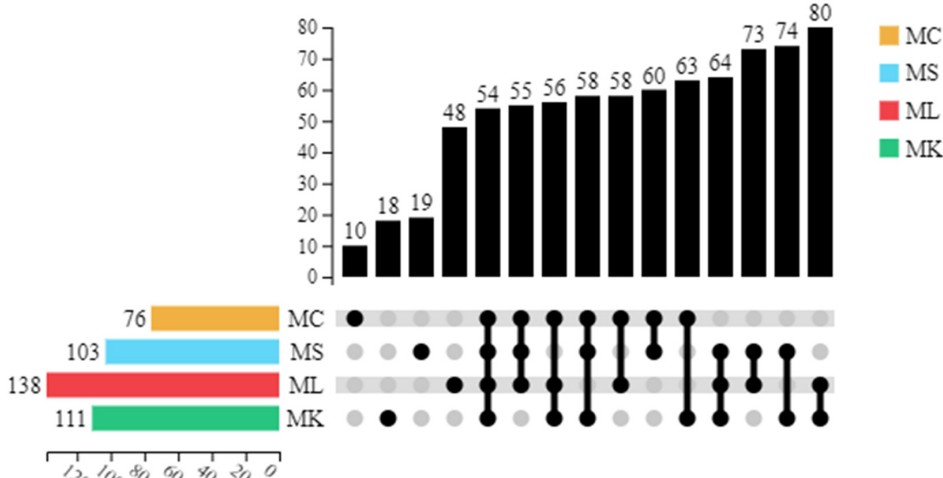

**Figure 1.** Upset Venn diagram of ECM in four types of plantation at the genus level. The bar graph on the left represents the element statistics of each group, a single dot in the middle matrix represents an element unique to a group, the line between the dots and the dot represents the unique intersection of different groups, and the vertical bar graph represents the corresponding the value of the intersection element.

**Table 3.** The relative abundance of ECM in the four types of *P. massoniana* plantations in spring.

| Name | | MC | MS | ML | MK |
|---|---|---|---|---|---|
| Phylum | Basidiomycota | 98 | 99.87 | 94.91 | 90.85 |
| | Ascomycota | 2 | 0.13 | 5.09 | 9.15 |
| Class | Agaricomycetes | 98 | 99.87 | 94.91 | 90.85 |
| | Dothideomycetes | 1.46 | 0.03 | 3.1 | 4.11 |
| | Pezizomycetes | 0.55 | 0.1 | 1.99 | 5.04 |
| Family | Russulaceae | 74.72 | 71.84 | 32.68 | 26.18 |
| | Thelephoraceae | 15.65 | 1.89 | 8.68 | 24.61 |
| | Clavulinaceae | 5.52 | 22.49 | 9.71 | 31.76 |
| | Gloniaceae | 1.46 | 0.03 | 3.1 | 4.11 |
| | Atheliaceae | 0.79 | 3.62 | 43.64 | 3.57 |
| | Inocybaceae | 0.72 | 0.02 | 0.18 | 4.73 |
| | Pyronemataceae | 0.55 | 0.1 | 1.99 | 5.04 |
| | Suillaceae | 0.55 | 0 | 0 | 0 |
| | Gomphidiaceae | 0.05 | 0 | 0 | 0 |
| | Amanitaceae | 0 | 0 | 0 | 0.01 |
| Genus | *Lactarius* | 45.31 | 30.06 | 10.05 | 0.44 |
| | *Russula* | 29.41 | 41.78 | 22.63 | 25.74 |
| | *Tomentella* | 15.65 | 1.89 | 8.68 | 24.61 |
| | *Clavulina* | 5.52 | 22.49 | 9.71 | 31.76 |
| | *Cenococcum* | 1.46 | 0.03 | 3.1 | 4.11 |
| | *Suillus* | 0.55 | 0 | 0 | 0 |
| | *Amphinema* | 0.44 | 3.45 | 0.19 | 3.39 |
| | *Tylospora* | 0.34 | 0.17 | 43.45 | 0.18 |
| | *Wilcoxina* | 0.04 | 0.1 | 0.12 | 4.73 |
| | *Geopora* | 0 | 0 | 0.48 | 0 |
| | *Inocybe* | 0.13 | 0 | 0 | 0 |
| | *Gomphidius* | 0.05 | 0 | 0 | 0 |
| | *Amanita* | 0 | 0 | 0 | 0.01 |
| | Unclassified | 1.1 | 0.02 | 1.59 | 5.04 |

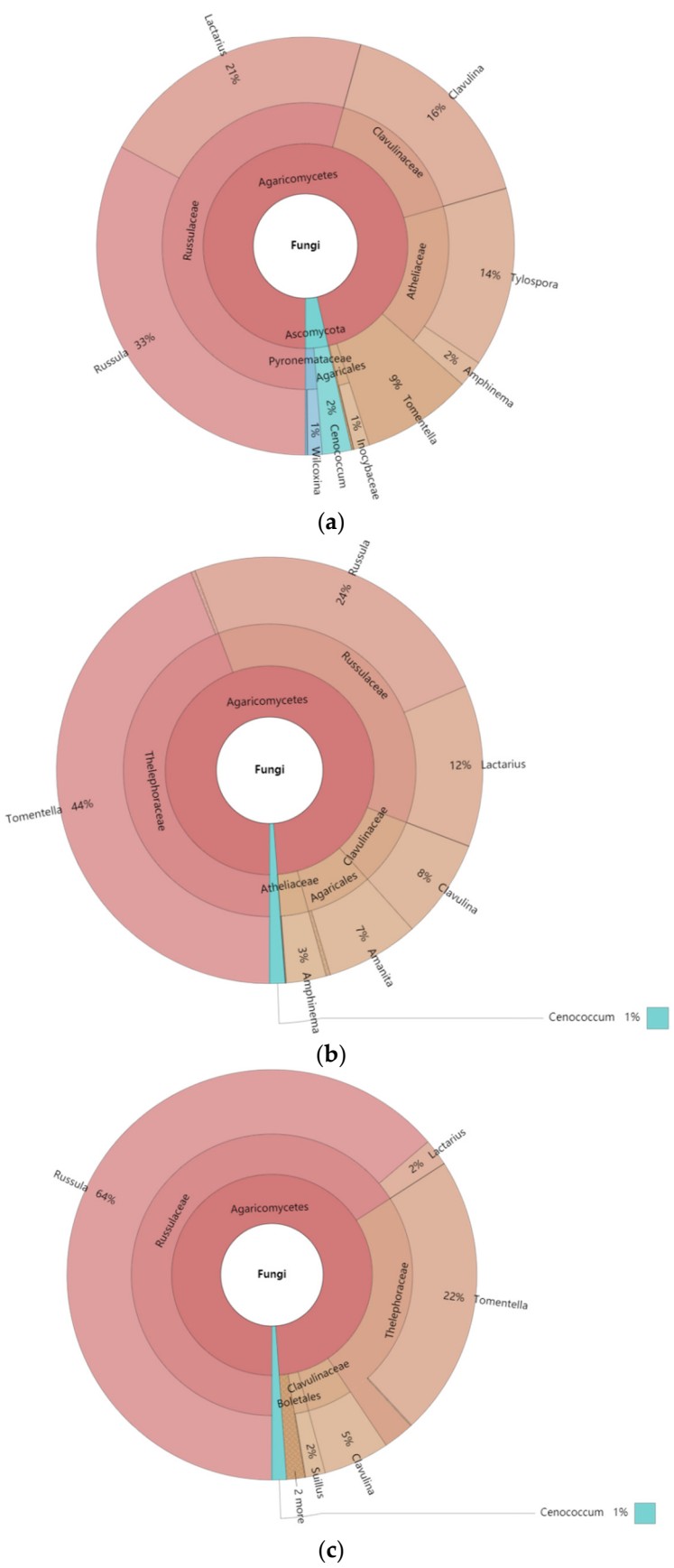

**Figure 2.** The species composition of ECM fungi in *Pinus massoniana* plantation. (**a**) Spring, (**b**) Summer and (**c**) Autumn. Different circles represent different levels.

In summer, the ECM fungi mainly came from two phyla, two classes, eight families and 12 genera, and Basidiomycota was the dominant phylum in the four forest types. At the class rank (Table 4), the ECM were mainly from Agaricomycetes and Dothideomycetes, in which Agaricomycetes was the dominant class. The dominant families were different in the four types of plots, and Hymenochaetaceae (0.02%) only existed in MS. Thelephoraceae was the dominant family in the four types of forests. The ECM in MC included 10 genera, of which the dominant genera were *Russula* (43.63%), *Amanita* (19.40%), *Lactarius* (19.04%), *Tomentella* (7.31%), *Clavulina* (5.25%) and *Amphinema* (4.76%). In MS, all the ECM came from 12 genera, and the dominant genera were *Russula* (42.11%), *Lactarius* (26.20%), *Clavulina* (11.89%), *Amphinema* (11.88%), *Tomentella* (3.71%), *Cenococcum* (1.80%), *Tomentellopsis* (1.32%), *Coltricia* (0.02%) and *Pseudotomentella* (0.01%). In ML, all the ECM came from eight genera, of which the dominant genera were *Tomentella* (33.36%), *Russula* (23.93%), *Clavulina* (21.13%), *Lactarius* (15.30%), *Cenococcum* (4.72%) and *Inocybe* (1.32%). *Tomentella* was the dominant genus in the four types of forest. *Russula*, *Lactarius* and *Clavulina* were the dominant genera in MC, ML and MK. *Amanita* was the dominant genus only in MC. *Inocybe* was the dominant genus only in ML. *Tomentellopsis* was the dominant genus only in MS.

**Table 4.** The relative abundance of ECM in the four types of *P. massoniana* plantations in summer.

| Name | | MC | MS | ML | MK |
|---|---|---|---|---|---|
| Phylum | Basidiomycota | 99.7 | 98.2 | 95.28 | 99.36 |
| | Ascomycota | 0.3 | 1.8 | 4.72 | 0.64 |
| Class | Agaricomycetes | 99.7 | 98.2 | 95.28 | 99.36 |
| | Dothideomycetes | 0.3 | 1.8 | 4.72 | 0.64 |
| Family | Russulaceae | 62.68 | 68.35 | 39.26 | 0.8 |
| | Thelephoraceae | 7.33 | 5.05 | 33.36 | 98.24 |
| | Clavulinaceae | 5.25 | 11.89 | 21.13 | 0.28 |
| | Amanitaceae | 19.4 | 0.17 | 0.16 | 0 |
| | Atheliaceae | 5.04 | 11.97 | 0.05 | 0.03 |
| | Gloniaceae | 0.3 | 1.8 | 4.72 | 0.64 |
| | Inocybaceae | 0.01 | 0.76 | 1.32 | 0.01 |
| | Hymenochaetaceae | 0 | 0.02 | 0 | 0 |
| Genus | *Tomentella* | 7.31 | 3.71 | 33.36 | 98.24 |
| | *Russula* | 43.63 | 42.11 | 23.93 | 0.63 |
| | *Lactarius* | 19.04 | 26.2 | 15.3 | 0.18 |
| | *Clavulina* | 5.25 | 11.89 | 21.13 | 0.28 |
| | *Amanita* | 19.4 | 0.17 | 0.16 | 0 |
| | *Amphinema* | 4.76 | 11.88 | 0.04 | 0.03 |
| | *Cenococcum* | 0.3 | 1.8 | 4.72 | 0.64 |
| | *Inocybe* | 0.01 | 0.76 | 1.32 | 0.01 |
| | *Tomentellopsis* | 0.01 | 1.32 | 0 | 0.01 |
| | *Tylospora* | 0.25 | 0.08 | 0 | 0 |
| | *Coltricia* | 0 | 0.02 | 0 | 0 |
| | *Pseudotomentella* | 0 | 0.01 | 0 | 0 |
| | Unclassified | 0.04 | 0.05 | 0.03 | 0 |

In autumn, all the ECM belonged to 2 phyla, 2 classes, 11 families and 14 genera. Basidiomycota was the dominant phylum in the four forest types (Table 5). At the class rank, Agaricomycetes was the dominant class, and the relative abundances in MC, MS, ML and MK were 99.98%, 96.81%, 99.93% and 83.78%, respectively. Gomphidiaceae existed only in the MS. There were six families in ML, which was the smallest number of the four types of forest. Russulaceae and Clavulinaceae were the dominant families. At the genus rank, *Russula* and *Clavulina* were the dominant genera.

**Table 5.** The relative abundance of ECM in the four types of *P. massoniana* plantations in autumn.

| Name | | MC | MS | ML | MK |
|---|---|---|---|---|---|
| Phylum | Basidiomycota | 99.98 | 96.81 | 99.93 | 83.78 |
| | Ascomycota | 0.02 | 3.19 | 0.07 | 16.22 |
| Class | Agaricomycetes | 99.98 | 96.81 | 99.93 | 83.78 |
| | Dothideomycetes | 0.02 | 3.19 | 0.07 | 16.22 |
| Family | Russulaceae | 93.39 | 70.91 | 13.75 | 33.51 |
| | Clavulinaceae | 6.06 | 1.68 | 24.96 | 16.17 |
| | Thelephoraceae | 0.47 | 16.19 | 58.45 | 24.82 |
| | Gloniaceae | 0.02 | 3.19 | 0.07 | 16.22 |
| | Amanitaceae | 0.02 | 0.13 | 0 | 0 |
| | Suillaceae | 0.02 | 7.58 | 0 | 0.08 |
| | Atheliaceae | 0.01 | 0.01 | 2.73 | 1.91 |
| | Gomphidiaceae | 0 | 0.31 | 0 | 0 |
| | Inocybaceae | 0 | 0.01 | 0.02 | 7.18 |
| | Boletaceae | 0 | 0 | 0 | 0.12 |
| Genus | *Russula* | 90.12 | 63.22 | 13.73 | 32.95 |
| | *Clavulina* | 6.06 | 1.68 | 24.96 | 16.17 |
| | *Lactarius* | 3.27 | 7.68 | 0.03 | 0.56 |
| | *Tomentella* | 0.4 | 1.14 | 57.88 | 24.46 |
| | *Cenococcum* | 0.02 | 3.19 | 0.07 | 16.22 |
| | *Suillus* | 0.02 | 7.58 | 0 | 0.08 |
| | *Amphinema* | 0 | 0.01 | 0.98 | 0.03 |
| | *Tylospora* | 0 | 0 | 1.75 | 1.87 |
| | *Tomentellopsis* | 0 | 0 | 0.38 | 0.03 |
| | *Inocybe* | 0 | 0.01 | 0.02 | 7.18 |
| | *Amanita* | 0.03 | 0.31 | 0 | 0 |
| | *Gomphidius* | 0 | 0.12 | 0 | 0 |
| | *Rhizopogon* | 0 | 0 | 0.01 | 0 |
| | *Tylopilus* | 0 | 0 | 0 | 0.13 |
| | Unclassified | 0.07 | 15.06 | 0.19 | 0.34 |

### 3.2. Alpha Diversity of ECM Fungi in P. massoniana

In spring, the Shannon and Simpson indices of the ECM fungi were the highest in MC, and the Chao1 and ACE indices were the highest in MS (Figure 3). One-way ANOVA showed that there was a significant difference in the Shannon index of ectomycorrhizal fungi among MC, ML, MS and MK ($p < 0.05$), with the order MC > ML > MS > MK. In summer, the Shannon index and Simpson index of ECM fungi were the highest in ML, and the indices of Chao1 and ACE were the highest in MS. The Chao1 and ACE indices of ECM fungi in MK were the lowest. In autumn, the Shannon and Simpson indices of ECM fungi were the highest in MK, and the Chao1 and ACE indices were the highest in MC. According to one-way ANOVA, the Shannon and Simpson indices of ECM fungi in MC were significantly different from those of the other three types ($p < 0.05$).

In MC, the Shannon and Simpson indices of ECM fungi were the highest in spring, and the Chao1 and ACE indices were the highest in autumn. In autumn, the Shannon and Simpson indices of ECM fungi were significantly lower than those in spring and summer ($p < 0.05$). In MS, the Shannon, Simpson, Chao1 and ACE indices of ECM fungi were all the highest in summer. The Shannon and Simpson indices of ECM fungi were significantly higher in summer than in autumn ($p < 0.05$). In MK, the Shannon and Simpson indices of ECM fungi were the highest in autumn, and the Chao1 and ACE indices were the highest in summer. The Shannon index of ECM fungi significantly differed from each other among the three seasons ($p < 0.05$).

The results showed that different sampling seasons had extremely significant effects on the Shannon, Simpson, Chao1 and ACE indices ($p < 0.01$) (Table 6), and different forest types had a significant effect on the Shannon, Simpson, Chao1 and ACE indices as well ($p < 0.01$). The interaction between sampling season and stand type had an extremely

significant influence on the Shannon and Simpson indices ($p < 0.01$) and a significant influence on the Chao1 and ACE indices ($p < 0.05$).

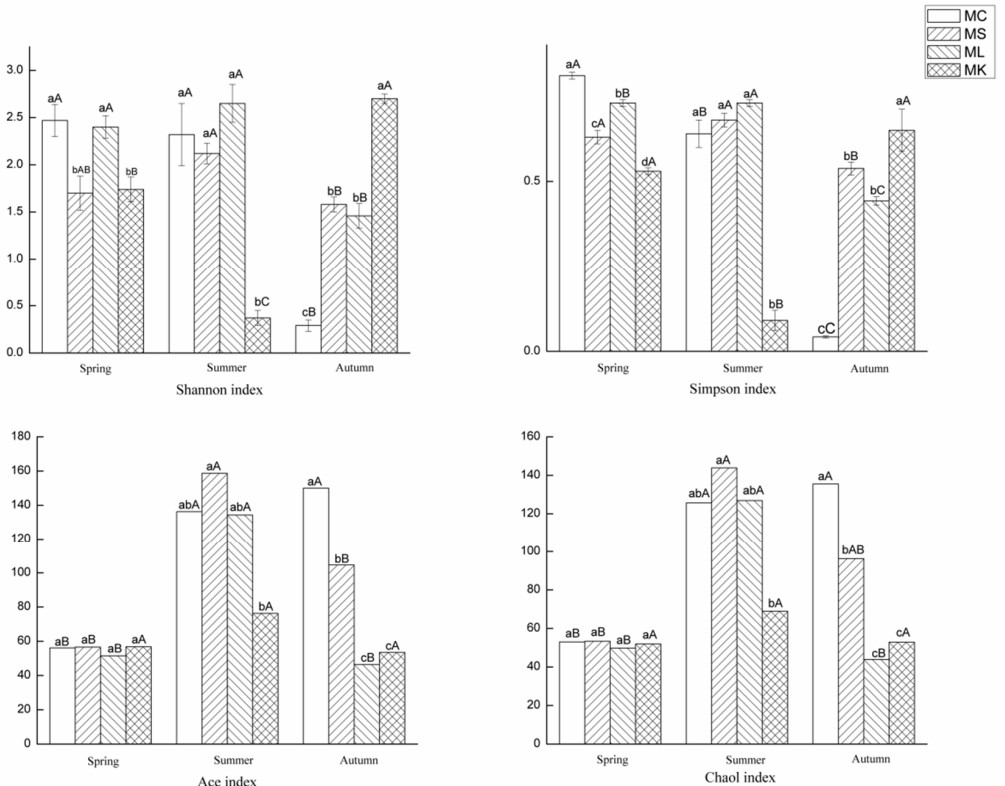

**Figure 3.** Alpha diversity index of ECM fungi in the four stand types of *Pinus massoniana* plantation. Values are the mean ± SE, and different lowercase letters indicate significant differences in the same stand types in different seasons at the $p < 0.05$ level; different uppercase letters indicate significant differences in different forest types in the same season at the $p < 0.05$ level.

**Table 6.** Results of two-way ANOVA for testing the main effects of forest type, season and their interactions on the alpha diversity index.

| Variable | Season | | | Forest Type | | | Season × Forest Type | | |
|---|---|---|---|---|---|---|---|---|---|
| | *df* | *F* | *P* | *df* | *F* | *P* | *df* | *F* | *P* |
| Shannon | 2 | 13.614 | 0 | 3 | 7.456 | 0.001 | 6 | 40.674 | 0 |
| Simpson | 2 | 99.241 | 0 | 3 | 52.965 | 0 | 6 | 94.173 | 0 |
| Chao1 | 2 | 24.721 | 0 | 3 | 8.424 | 0.001 | 6 | 4.327 | 0.004 |
| ACE | 2 | 27.386 | 0 | 3 | 9.679 | 0 | 6 | 4.857 | 0.002 |

*3.3. Principal Coordinate Analysis of ECM Fungal Community in P. massoniana*

The results showed that the ECM fungal communities of the four types of plantations were different, and the samples in the same stand could gather together (Figure 4a). The ECM fungal samples in MC were relatively close to MS, indicating that the ECM fungal community structure of these two was relatively similar. The distance among ECM fungal samples in ML with the other three was far, it shows that the ECM fungal community structure of ML was less similar to the other three stand types. The results of Permanova test showed that ECM fungal community composition was significantly different among the four types of plantations ($p = 0.001$). In summer, the contribution rates of the first and second principal components were 18.27% and 12.76%, respectively, which was explain 29.03% of all differences (Figure 4b). The ECM fungal communities of MC and MS are less similar. The results of Permanova test showed that ECM fungal community composition

was not significantly different among the four types of plantations ($p > 0.05$). Therefore, in autumn (Figure 4c), The ECM fungal communities of the four types of forests were different, and the samples in the same forest can be clustered together. Only MC and MS showed a little similarity, and there were significant differences with the other two communities. In summary, the treatments were clearly separated in spring and summer.

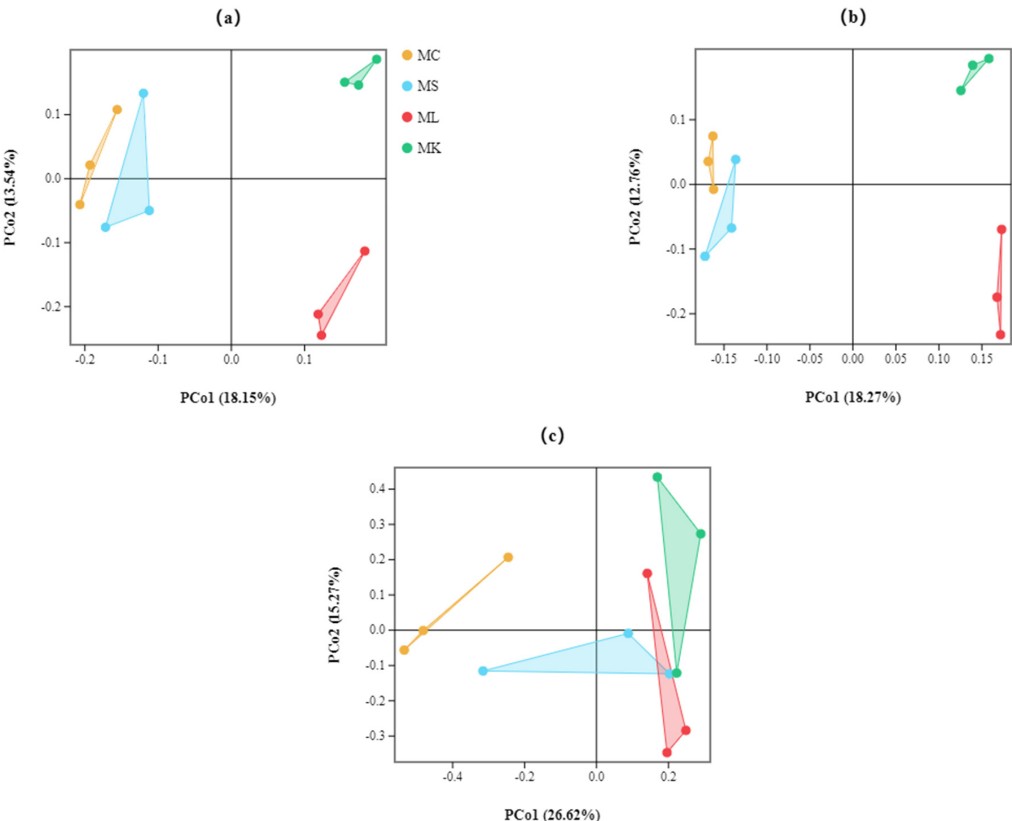

**Figure 4.** The PCoA analysis of ECM fungi community in four types of *Pinus massonian* plantation. (**a**) Spring, (**b**) Summer and (**c**) Autumn.

### 3.4. Effects of Soil Physical and Chemical Properties on ECM in P. massoniana

In the three seasons, SOM, C/N and TP were all highest in MK (Table 7). In spring, the water content, TN and TK in ML were highest, and pH was highest in MS. In summer, the water content (SM) and TN in ML were highest, the pH was highest in MS, and the TK in MK was highest. In autumn, the water content in MC was highest, the pH in ML was highest, the TN in MS was highest and the TK in MK was highest.

Detrended correspondence analysis (DCA) was performed on the ECM fungal species data matrix of the four forest types in the three seasons, and the results showed that the lengths of gradient were all less than 3.0; therefore, the linear model (RDA) was selected to analyze the effect of water content, pH, SOM, total nitrogen (TN), C/N, TP and TK on ECM.

In spring, the total explanation rate of all factors was 91.50% (Figure 5, Spring). The effects of soil physical and chemical properties on the ECM genera of the four types of forest were in the following order: TN > TP > SOM > C/N > pH > TK > SM. TN, TP, SOM and C/N had significant effects on the ECM fungal genera ($p < 0.05$). There was an obvious synergistic effect between water content and TK, which was mainly positively correlated with *Tylospora*, *Wilcoxina* and *Geopora* and negatively correlated with *Lactarius*, *Suillus* and *Gomphidius*. In summer, the total explanation rate of all factors was 92.56% (Figure 5, Summer). The effects of soil physical and chemical properties on the ECM genera of the four types of forest were in the following order: TP > SOM > pH > C/N > TN >

TK > SM. TP and SOM had significant effects on the ECM fungi ($p < 0.05$). In autumn, the total explanation rate of all factors was 94.27% (Figure 5, Autumn). The effects of soil physical and chemical properties on the ECM genera of the four types of forest were in the following order: pH > SM > SOM > TP > TK > C/N > TN; among them, pH showed a significant effect on the ECM fungal composition ($p < 0.05$).

**Table 7.** The soil physical and chemical properties in four types of *Pinus massonian* plantation in spring, summer and autumn.

| Season | Types | SM | pH | SOM (g·kg⁻¹) | TN (g·kg⁻¹) | C/N | TP (g·kg⁻¹) | TK (g·kg⁻¹) |
|---|---|---|---|---|---|---|---|---|
| Spring | MC | 35.03 ± 1.19 ab | 3.64 ± 0.11 b | 38.10 ± 0.67 b | 1.96 ± 0.04 a | 19.47 ± 0.61 b | 2.74 ± 0.03 d | 11.71 ± 0.12 b |
| | MS | 32.61 ± 0.95 b | 4.36 ± 0.07 a | 37.17 ± 1.26 b | 1.77 ± 0.01 b | 20.99 ± 0.83 b | 3.16 ± 0.01 c | 11.34 ± 0.15 b |
| | ML | 37.43 ± 0.40 a | 3.69 ± 0.07 b | 31.34 ± 0.73 c | 2.02 ± 0.02 a | 15.49 ± 0.44 c | 3.43 ± 0.03 b | 12.52 ± 0.11 a |
| | MK | 36.74 ± 0.80 a | 3.62 ± 0.04 b | 53.66 ± 1.38 a | 1.46 ± 0.05 c | 36.98 ± 0.60 a | 4.45 ± 0.06 a | 11.65 ± 0.16 b |
| Summer | MC | 38.46 ± 0.41 a | 4.09 ± 0.04 b | 32.79 ± 0.88 d | 2.61 ± 0.06 a | 12.59 ± 0.37 c | 2.70 ± 0.01 c | 13.37 ± 0.18 b |
| | MS | 38.30 ± 1.03 a | 4.94 ± 0.11 a | 38.09 ± 0.46 c | 2.32 ± 0.05 b | 16.46 ± 0.48 b | 2.34 ± 0.06 d | 12.18 ± 0.10 d |
| | ML | 39.37 ± 0.38 a | 3.88 ± 0.05 b | 48.89 ± 0.82 b | 2.65 ± 0.04 a | 18.44 ± 0.45 b | 3.03 ± 0.03 b | 12.85 ± 0.07 c |
| | MK | 37.73 ± 1.27 a | 3.87 ± 0.05 b | 51.92 ± 0.94 a | 1.60 ± 0.13 c | 32.73 ± 1.99 a | 3.96 ± 0.03 a | 14.17 ± 0.06 a |
| Autumn | MC | 38.93 ± 0.40 a | 3.96 ± 0.12 b | 36.31 ± 0.52 b | 1.63 ± 0.04 a | 22.34 ± 0.52 b | 2.48 ± 0.03 c | 16.40 ± 0.03 a |
| | MS | 38.29 ± 0.37 a | 3.93 ± 0.09 b | 34.58 ± 0.64 b | 1.64 ± 0.08 a | 21.16 ± 1.33 b | 3.56 ± 0.05 b | 14.35 ± 0.21 c |
| | ML | 34.79 ± 1.02 b | 4.31 ± 0.14 a | 32.57 ± 0.68 c | 1.50 ± 0.09 a | 21.90 ± 1.53 b | 3.76 ± 0.14 b | 15.71 ± 0.13 b |
| | MK | 36.45 ± 1.06 ab | 3.86 ± 0.05 b | 51.76 ± 0.57 a | 1.45 ± 0.08 a | 36.02 ± 2.24 a | 4.50 ± 0.08 a | 16.34 ± 0.12 a |

Values are the mean ± SD. Different lowercase letters indicate significant differences among different stand types within the same season at $p < 0.05$.

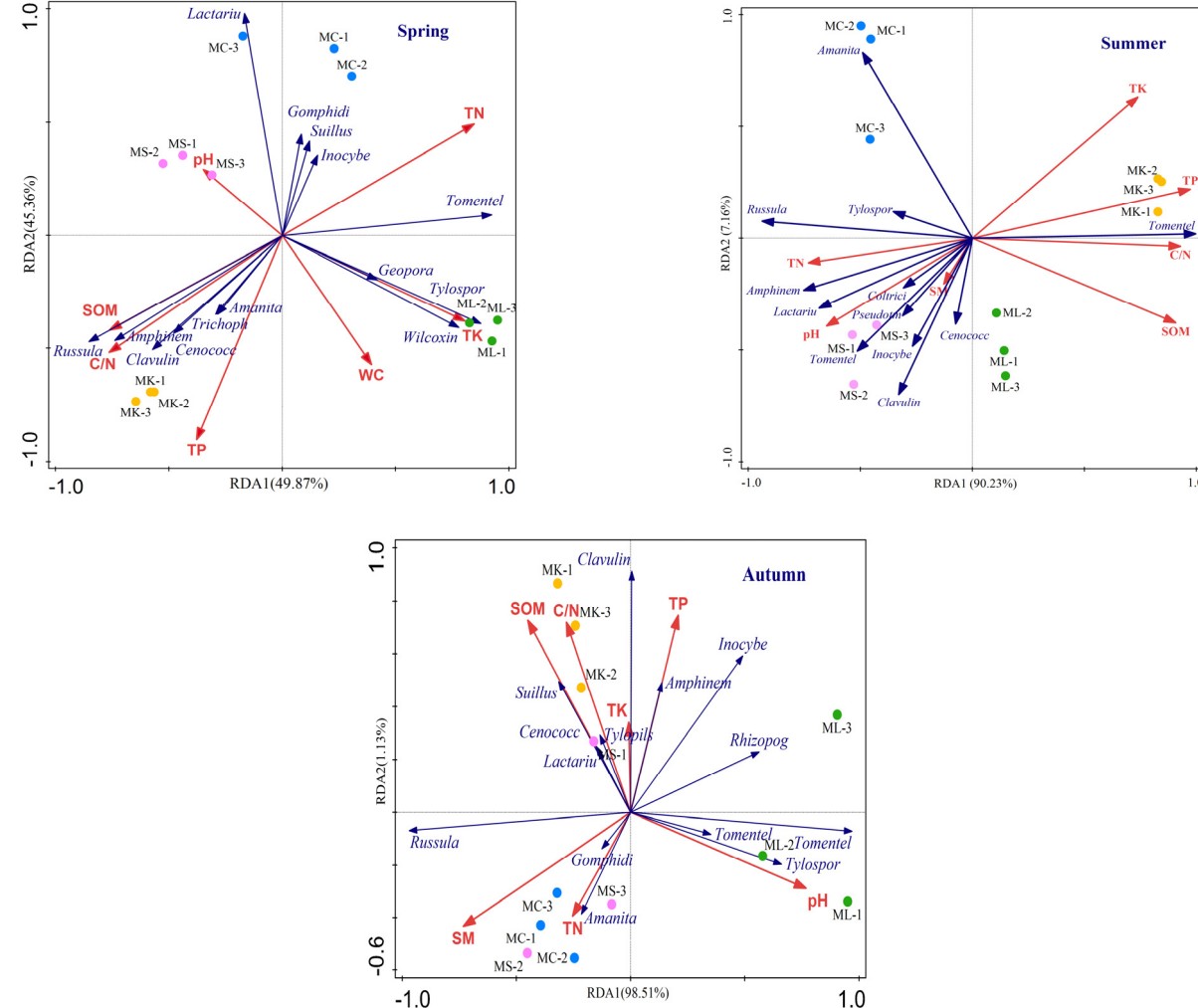

**Figure 5.** RDA of the ECM fungal community and soil physical and chemical properties in the four types of *P. massoniana* plantations in the three seasons.

## 4. Discussion

### 4.1. Composition of ECM Fungi in the Four Types of P. massoniana Forest

The dominant class of ECM fungi in the three seasons was Agaricomycetes, accounting for 53% of Basidiomycota, which has the most important ectomycorrhizal symbiosis pattern in forests [26]. According to statistics, Russulaceae is one of the dominant families of ECM fungi in southern China [27]. Russulaceae occur on the ground in forests, rarely on rotting wood, usually forming mycorrhizae with higher plants and are distributed almost all over the world [28]. *Russula* and *Lactarius* are both important ECM fungi in Russulaceae [29]. *Russula* is a type of large umbrella-shaped fungus with a wide range of species in the world, most of which can form ectomycorrhizae with a variety of plants [30], which has high economic and research value; however, some species are poisonous. *Lactarius* is one of the most important specific ECM fungal genera on Earth [31] and forms ectomycorrhizae with wide range of plant species. A large number of studies have found that ECM fungi of *Tomentella* are distributed globally and are not host specific. They can form ECM symbioses with a variety of plants and are an important part of the global ECM fungal community [32–35]. *Tomentella* is rich in species because of its rich symbiosis-forming strategies, and the ECM fungi of this genus have a strong adaptability to the environment and strong colonization ability [36,37].

Clavulina, a common ECM genus, has no specific hosts and is widely distributed [38]. *Cenococcum geophilum* is distributed globally and has a wide range of hosts. *Pinus massoniana*, *Keteleeria fortunei* and *Pinus sylvestris* can symbiotically form ectomycorrhizae with *Cenococcum geophilum* [39]. *Cenococcum* is one of the few genera that can be identified directly by morphological characteristics. *Cenococcum geophilum* mostly exists in arid-soil environments and shows strong resistance to environmental stress, especially drought, because of its large amount of melanin [40,41].

ECM fungi play an indispensable role in a stable forest ecosystem. The fungi involved in ecosystem processes are not foreign but have resulted from a common evolution with the ecosystem. In this study, the four types of *P. massoniana* plantations were in a relatively stable state. Different mixing patterns changed the microenvironment in the stands, and the competitiveness of the trees against various resources was different, which makes the growth of the root systems different, including synergy or competition between ECM. As a result of natural selection, a stable and efficient ECM community was formed.

### 4.2. Alpha Diversity of ECM Fungi

The analysis of the results showed that the sampling season had extremely significant effects on the alpha diversity index of the ECM fungal communities for the same forest type. There were certain differences in the species composition and relative abundance of ECM in different sampling seasons, which contributed to alpha diversity. Climate is one of the abiotic factors that affects the composition of the ECM community, especially rainfall and temperature [27,35]. Elevated temperature increases the number of ECM that have higher biomass and proteolytic ability (especially *Cortinarius* spp.) and decreases the number of ECM fungi that have an affinity for unstable nitrogen (especially *Russula* spp.). The study area is located at the eastern edge of the Sichuan Basin, a subtropical humid monsoon climate zone, with climate characteristics that include drought in the spring, heat in the summer, cool weather in the autumn, warmth in the winter, abundant rainfall, four distinct seasons and sufficient sunshine. Rainfall and temperature differ distinctly in different seasons; therefore, they directly or indirectly affected the composition of the ECM community, which also significantly affected the alpha diversity of the ECM. During the same season, different forest types also had a significant impact on the alpha diversity index of the ECM communities. There were certain differences in the species composition and relative abundance of ECM fungi of different forest types, leading to differences in alpha diversity. Biological factors are likely to be one of the main causes of this difference. The biological factors that affect the composition of the ECM community mainly include the type of host, the diameter and age of the host, the type and diversity of other plants in

the plots, soil fauna and other microorganisms [42]. Changes in the species and diversity of aboveground plants affect the use of space and resources of the host plant and have certain impacts on the production of host plant root exudates. *Cunninghamia lanceolata*, *Cryptomeria fortunei* and *Cinnamomum camphora* can all form arbuscular mycorrhizae with AM fungi [43]. The addition of these arbuscular mycorrhizal tree species changed the utilization of the soil fungal bank by the original tree species and may have changed the original ECM fungal community structure. Therefore, the diversity of ECM fungi in the four types of forest was significantly influenced by the mixed types.

*4.3. Correlation between Soil Physical and Chemical Properties and the ECM Communities*

The life history of ECM fungi is completely in the soil; thus, the physical and chemical properties of the soil directly affect the diversity of the ECM fungi [14]. Soil pH affects the growth and metabolism of mycelium of the ECM fungi. In general, partially acidic soil, especially with a pH range from 4 to 6, is suitable for ECM fungi [44]. Similarly, the *P. massoniana* forest is not tolerant to saline-alkali, acidophilic and slightly acidic soil, and its pH value is 4.5–6.5. ECM fungi and *P. massoniana* have the same preference in soil pH, which indicates that the adaptability of *P. massoniana* to the acidic soil environment is likely to be related to the ECM fungi [45]. Most ECM fungi form mycorrhizal symbioses in the presence of abundant litter and rich organic matter. Therefore, the content of organic matter affects the formation of most ECM fungi. The study found that the enzymatic activity of ECM-surrounding soil to promote the decomposition of complex organic matter in animal and plant residues was significantly higher than that in nonmycorrhizal soil. This indicates that ECM fungi have the ability to decompose soil organic matter and to participate in the forest carbon cycle [46,47]. ECM fungi promote the absorption of nutrients by the host-plant roots after forming ectomycorrhizae symbiotically [48]. Both nitrogen and phosphorus are important nutrients required by plants, and they are also important factors affecting the symbiotic relationship between the ECM and host plants [49]. In this study, the soil TN and TP contents were both low, but the trees maintained normal growth, indicating that ECM fungi played an important role in promoting the growth of the host plants in this area. Host plants absorb inorganic N in the soil directly, while some ECM fungi absorb organic N and secrete nitrate reductase to transform inorganic N into organic N. In the case of low P content, to improve their own nutritional status, the host plant uses ECM fungi to absorb more P [50], and the number of ECM fungi that have the ability to promote phosphorus absorption may be changed; thus, the community structure of ECM fungi changes simultaneously. After the ECM fungus and the host plant form a symbiosis, the P enriched by the root epidermis and mycorrhiza is absorbed and utilized by multiple phosphorus transporter genes of the same or different families [42], of which Pht1 plays an important role that is responsible for the absorption and transport of P in plants, belonging to the MFS protein family [51]. Plants can use the osmotic potential gradient to transport solute molecules, and it has been found that most Pht1 phosphorus transporter gene expression is induced by low P content [52,53].

**5. Conclusions**

As a near-natural forest management model, the mixed forest is of great significance for improving the quality of plantations. This study found that the colonization rate of ECM in MS was the highest in the four forest types. In addition, different seasons also significantly affected the composition of the ECM community, among which the diversity of ECM was highest in the summer. The physical and chemical properties of the soil were also one of the key factors affecting the diversity of the ECM, especially TP, soil pH and SOM. In the future, it will be particularly important to explore the relationship between ectomycorrhizae and environmental changes on a larger scale and from more perspectives.

**Author Contributions:** Conceptualization: X.L. (Xiangjun Li) and W.K.; methodology, S.L.; software, G.C.; validation, X.L. (Xianwei Li), J.L. (Jiangli Liu) and Q.L.; formal analysis, K.Z.; investigation, X.L. (Xiangjun Li); resources, X.L.(Xianwei Li); data curation, J.L. (Junjie Liu); writing—original draft preparation, X.L. (Xiangjun Li); writing—review and editing, H.Y. and C.F.; visualization, G.C.; supervision, X.L. (Xiangjun Li); project administration, Y.S.; funding acquisition, X.L. (Xianwei Li). All authors have read and agreed to the published version of the manuscript.

**Funding:** This study was supported by the National Key Research and Development Program of China (Grant No. 2017YFD060030205), the German Government loans for Sichuan. Forestry Sustainable Management (Grant No. G1403083), and the "Tianfu Ten Thousand Talents Plan" of Sichuan Province (Grant No. 1922999002).

**Acknowledgments:** This study was supported by the project above. We also thank all professors who provided helpful guidance in this research.

**Conflicts of Interest:** The authors declare no competing interest.

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
