# Peer review of "Diversity of Ectomycorrhizal Fungal Communities in Four Types of Stands in Pinus massoniana Plantation in the West of China"

_forests, doi:10.3390/f12060719_

Round 1
Reviewer 1 Report
The authors have explored the diversity of ECM in different Pinus species in a plantation from 1980 in Western China. Besides analysing standard soil parameters, the fungal communities were analysed based on the ITS2 gene segment by Illumina sequencing technologies. The sequence data were then evaluated by different statistical methods.
The results obtained provide essential novel data on the diversity of fungal communities in the studied area. However, since a classical approach was used (comparing soil chemistry data with the fungal diversity), the study is bound to find evidence of at least some kind of correlations with soil chemistries and organic matter. This is of course expected, but to reach a deeper understanding, it may be wise to include the impact of recent discoveries, such as the impact of prokarytoes and other microbes in the soil, as these may also influence the fungal diversity (e.g. as helper bacteria (often abbreviated as MHB). The authors are not requested to sequence the prokaryotes for this study, but I would recommend you to include the potential role of MHB in your discussion (and if possible, also in the introduction).
Question to 2.1 study area: what about the presence of animals - and their impact?; and was any kind of managment performed of the study area (fertilization or other measures?)
Author Response
Thanks for your suggestions,
Point 1: Question to 2.1 study area: what about the presence of animals - and their impact?; and was any kind of managment performed of the study area (fertilization or other measures?)
Response 1: The study site is a state-owned forest farm with no large animals lived. The forest had the necessary forest management measures, including weeding and removing dead wood, but no chemical measures, such as fertilization. I've add it in my manuscript.
Point 2: The results obtained provide essential novel data on the diversity of fungal communities in the studied area. However, since a classical approach was used (comparing soil chemistry data with the fungal diversity), the study is bound to find evidence of at least some kind of correlations with soil chemistries and organic matter. This is of course expected, but to reach a deeper understanding, it may be wise to include the impact of recent discoveries, such as the impact of prokarytoes and other microbes in the soil, as these may also influence the fungal diversity (e.g. as helper bacteria (often abbreviated as MHB). The authors are not requested to sequence the prokaryotes for this study, but I would recommend you to include the potential role of MHB in your discussion (and if possible, also in the introduction).
Response 2:The study site is a state-owned forest farm with no large animals lived. The forest had the necessary forest management measures, including weeding and removing dead wood, but no chemical measures, such as fertilization. I've add it in my manuscript.

Reviewer 2 Report
The presented work is an important contribution to the issue of synergistic research ECM fungi - mixed forests - diversity - chemical elements in the soil - pH - humidity, seasonality and others in relation to forest management. The authors had to carry out a number of field and demanding laboratory work. Good research design was supported by results obtained by demanding molecular genetic methods.
However, I must remind you that corrections to the References will be necessary! My comments and corrections I put directly into the text of the article.

Author Response
Thanks for your suggestions, I've revised my manuscript according to your comments and corrections.
Point 1: Pay attention to the way of quoting works in the text of the article and also in References ... 1) according to the instructions for authors in Forests - see www.mdpi.com/journal/forests/instruction ... where citations are quoted in a way, e.g. [1, 2] or [10-15] or 2 ....) are relied on your way.
Response 1:Yes, I’ve revised my manuscript according to the instructions for authors in Forests.
Point 2: this citation is missing in References, line 50 and .....
Response 2:I’ve add all references those I missed.
Point 3:A reference to Table 1 is missing in the text of the article.
Response 3:Yes, table 1 must be mentioned in study area, I’ve revised it.
All other comments from reviewer 2 have been revised in my manuscript, thanks a lot.

Reviewer 3 Report
Review of the manuscript "Diversity of ectomycorrhizal fungal communities in four types of Pinus massoniana plantation in the west of China".
My decision on the paper is between "reject" and "major revision". From the abstract on the text is full of typographical and grammatical errors, some sentences do not make sense, other sentences are not completed etc. Some thoughts are not supported by citations/references. Introduction part should be enriched with more text which describe past research on fungal communities of P. massoniana. Or if those type of research is not performed before, the the authors should refer to similar research on other pine or coniferous species in China. The English used is in the manuscript is mostly of a very low quality and I had problems to understand what are authors trying to say in some parts of the text. Results and Discussion sections are much better written than the abstract, Introduction or Materials and methods part.
So I recommend that the authors thoroughly check, clarify and revise the text. If they do not have enough knowledge of English, my advice for them is to use English Editing Service (from MDPI or elsewhere).
Please see the pdf of the manuscript attached with all my remarks, corrections, and questions.
Best wishes, Reviewer

Author Response
Thanks for your review report.
Point 1:unfinished sentence- line 15.
Response 1: I’ve revised it, As a broad-spectrum, symbiotic tree species of ECM, Pinus massoniana forms symbioses with various ECM fungi to form mycorrhizae.
Point 2: what kind of significant impact? - line 25
Response 2: I’ve revised this sentence, (In summary, there were significantly differences in ECM fungal communities among different forests types and different seasons).
Point 3:Question: which kind of transformation? State briefly.-line 26
Response 3:I’m sorry that the word was wrong, the new sentence as follow, Diversity of ECM fungal communities in P. massoniana-Cunninghamia lanceolata was the most significant.
Point 4: Question: What about plants, mosses etc.? line 38
Response 4: Yes, of course, I’ve add it in the text.
Point 5: "mycosis" is unadequate word- line 49
Response 5: I think ring traps will be better.
Point 6: word does not exist in English dictionary-line 54
Response 6: non-mycorrhizalized, means not mycorrhizalized, such as crop tree and noncrop tree.
Point 7: What do you mean by Mycorrhizal samples? Mycorrhizal root tips or soil samples? It is very important, so please clarify very carefully.line 99
Response 7: It means mycorrhizal root tips, I’ve revised it.
Point 8: 16S for mycorrhizal fungi? Why you did not use ITS (or ITS1 or ITS2) which gives much better resolution?
Response 8: Yes, you’re right, it’s my fought, we use ITS2 for this study, and the primers are ITS3_KYO2 and ITS4.
Point 9: How did you delimit fungal species based on trophic status (ECM fungi from saprotrophic/parasites)? Please describe in Materials and methods- line 185
Response 9: I’m sorry to miss it, I’ve added it in 2.2 Sample Collection and Processing.
I’ve revised my manuscript according to reviewer’s comments, include grammar mistakes, spell mistakes and errors in preference, please check it, thanks very much.

Round 2
Reviewer 3 Report
Dear authors and editors,
Please find the second round of review attached (as pdf file). The paper is now suitable for publication after minor corrections.
Best,
Reviewer

Author Response
Point 1: Diversity or also colonization rate?-line 27
Response 1: Yes,it means colonization rate, I’ve revised it.
Point 2:what does it mean? Diversity of ECM was the highest? -line 28
Response 2:Yes, I also think the sentence is not correct, so I revised it as follow. The Colonization rate of ECM fungal in P. massoniana-Cunninghamia lanceolata was the highest, so we infer that Cunninghamia lanceolata is the most suitable tree species for mixed with P. massoniana in three mixture forests.
I’ve revised my manuscript according to reviewer’s comments, include grammar mistakes, spell mistakes and errors in preference, please check it, thanks very much.
